# Alveolar Ridge Augmentation Assessment Using a Minimalistic Approach, with and without Low-Level Laser Therapy (LLLT)—A Comparative Clinical Trial

**DOI:** 10.3390/medicina59061178

**Published:** 2023-06-20

**Authors:** K. Padmanabhan Akhil, Rashmi Pramashivaiah, Munivenkatappa Laxmaiah Venkatesh Prabhuji, Robina Tasleem, Hussain Almubarak, Ghadah Khaled Bahamdan, Alexander Maniangat Luke, Krishna Prasad Shetty, Niher Tabassum Snigdha, Shaeesta Khaleelahmed Bhavikatti

**Affiliations:** 1Department of Periodontology, Krishnadevaraya College of Dental Sciences, Rajiv Gandhi University of Health Sciences, Bangalore 562157, India; drakhilkp@gmail.com (K.P.A.); drrashmip.perio@kcdsh.org (R.P.); drprabhuji.perio@kcdsh.org (M.L.V.P.); 2Department of Prosthodontics, College of Dentistry, King Khalid University, Abha 62529, Saudi Arabia; rkhattak@kku.edu.sa; 3Department of Diagnostic Sciences & Oral Biology, College of Dentistry, King Khalid University, Abha 62529, Saudi Arabia; hualmubarak@kku.edu.sa; 4Department of Periodontics and Community Dental Sciences, College of Dentistry, King Khalid University, Abha 62529, Saudi Arabia; ghbahamdan@kku.edu.sa; 5Department of Clinical Science, College of Dentistry, Ajman University, Al-Jurf, Ajman 346, United Arab Emirates; 6Centre of Medical and Bio-allied Health Sciences Research, Ajman University, Al-Jurf, Ajman 346, United Arab Emirates; kprasad11@gmail.com; 7Pediatric Dentistry Unit, School of Dental Sciences, Universiti Sains Malaysia, Health Campus, Kubang Kerian 16150, Kelantan, Malaysia; drnihertabassum@gmail.com; 8Department of Periodontics, School of Dental Sciences, Universiti Sains Malaysia, Health Campus, Kubang Kerian 16150, Kelantan, Malaysia; 9Center for Transdisciplinary Research (CFTR), Saveetha Dental College, Saveetha Institute of Medical and Technical Sciences, Saveetha University, Chennai 600077, India

**Keywords:** alveolar ridge, bone graft, low-level laser therapy, minimalistic, ridge augmentation, tunneling

## Abstract

*Background and Objective*: A narrow alveolar ridge is an obstacle to optimal rehabilitation of the dentition. There are several complex and invasive techniques to counter the ridge augmentation dilemma, with most of them exhibiting low feasibility. Hence, this randomized clinical trial aims to evaluate the effectiveness of a Minimalistic Ridge Augmentation (MRA) technique, in conjunction with low-level laser therapy (LLLT). *Materials and Methods*: A total of 20 patients (*n* = 20) were selected, with 10 assigned to the test group (MRA+LLLT) and the other 10 to the control group (MRA). A vertical incision of approximately 10 mm was placed mesial to the defect and tunneled to create a subperiosteal pouch across the entire width of the defect. At the test sites, a diode laser (AnARC Fox^TM^ Surgical Laser 810 nm) was used to deliver LLLT (parameters: 100 mW, with a maximum energy distribution of 6 J/cm^2^ in the continuous wave mode for 60 s per point) to the exposed bone surface inside the pouch, followed by graft (G-Graft, Surgiwear^TM^, Shahjahanpur, India) deposition with a bone graft carrier. The control sites were not irradiated with a laser. *Results*: A horizontal ridge width gain of >2 mm was observed in both groups. The changes in bone density for the test and control groups were −136 ± 236.08 HU and −44.30 ± 180.89 HU, respectively. Furthermore, there was no statistically significant difference between the test and control groups in these parameters. *Conclusion*: The study findings reveal that the MRA technique is relatively simple and feasible for alveolar ridge augmentation. The role of LLLT in the process requires further elucidation.

## 1. Introduction

Residual alveolar ridge resorption following tooth loss is inevitable. Extraction of the tooth due to periodontitis often results in significant loss of bone volume, both vertically and horizontally [1]. Adequate bone volume is the main prerequisite for functional and aesthetic rehabilitation after dental implants [2,3]. Alveolar ridge defects have been conventionally treated with several techniques that involve reflection of a mucoperiosteal flap, such as interposition grafting, ridge splitting, alveolar distraction osteogenesis, and cortico-cancellous block onlay grafting [4,5,6,7,8]. Autogenous bone has traditionally been used to reconstruct ridge defects and is considered to be the gold standard of treatments [1].

The obvious drawbacks inherent in this method include the need for an extra donor site, extensive surgery, along with its attendant complications, and reduced patient compliance compared to minimally invasive procedures [9,10]. In addition to the invasiveness of these procedures, they may also cause complete or incomplete graft loss due to graft exposure and postoperative wound dehiscence, particularly with onlay graft techniques that follow graft resorption through the remodeling phase of the cortico-cancellous block grafts [9,11,12,13,14].

Minimally invasive procedures have been shown to exhibit improved patient acceptance, as well as decrease intra-operative bleeding, postoperative pain, and swelling, while conserving and/or enhancing the soft tissue profiles. The subperiosteal tunneling technique requires constructing a subperiosteal flap with a small vertical incision, forming a pocket for the retention of graft materials without the need for a membrane [10].

Low-level laser therapy (LLLT) has been found to facilitate bone repair processes [15] and to reduce postoperative pain and other complications [16,17]. LLLT is proven to promote bone healing at bone fracture sites, and more importantly, extraction sockets; when given at the proper doses and frequency, LLLT promotes the differentiation and proliferation of human osteoblasts in culture when compared to the results of non-irradiated cells [18,19,20]. However, there have been limited reports in the literature regarding the benefit of adjunctive LLLT in ridge augmentation procedures.

In this study, a novel minimalistic ridge augmentation technique (MRA) is performed in combination with LLLT to optimize the outcome of the minimally invasive procedure.

## 2. Materials and Methods

### 2.1. Study Design and Patients

This study was a prospective, randomized, and controlled clinical trial. Patients were recruited from the outpatient department of the Department of Periodontology, Krishnadevaraya College of Dental Sciences, Bengaluru.

Patients over the age of 18 exhibiting a keratinized tissue width greater than 1.5 mm, Siebert’s class I and II defects corresponding to Cologne’s classification h1e and h2e defects, and with a span of not more than 25 mm, were included in the study. Patients with poor oral hygiene, thin biotypes (<0.8 mm), systemic diseases such as uncontrolled diabetes mellitus or hemorrhagic disorders, bone disorders such as osteoporosis, and who are taking immunosuppressive drugs were excluded from the study.

A total of 20 healthy adults aged between 21 and 60 years and exhibiting inadequate alveolar ridge dimensions in the maxillary and mandibular regions were randomized into test (*n* = 10) and control (*n* = 10) groups. The test group received minimalistic alveolar ridge augmentation with LLLT, and the control group received minimalistic ridge augmentation alone. The primary objective of the study was to measure the gain in ridge width (GRW) and assess the bone quality of the regenerated bone in Hounsfield units (HU). The secondary objective of the study was to explore the adjunctive role of LLLT in minimally invasive ridge augmentation. Other parameters measured were patient-related outcomes (PRO) such as pain, swelling, and discomfort during and after the procedure. A flow chart of the study design is presented in Figure 1.

The study was performed in compliance with the ethical guidelines outlined in the World Medical Association Declaration of Helsinki (version VI, 2002). Ethical approval was obtained from the Ethical Committee of Krishnadevaraya College of Dental Sciences (affiliated with Rajiv Gandhi University of Health Sciences). Furthermore, all participants were apprised of the nature of the study, as well as the potential benefits and risks of their participation. All participants provided informed written consent.

### 2.2. Clinical Measurement, Radiological Assessment, and Pre-Surgical Procedures

A preoperative clinical examination was carried out before the procedure. Study casts were prepared with dental stones for all the treated cases to plan the surgical procedure and estimate the amount of augmentation required.

The clinical ridge height and width were measured preoperatively using a University of North Carolina 15* (UNC-15) periodontal probe. Radiographic measurements (were performed using Cone-Beam Computed Tomography (CBCT) and Newtom CBCT 3D imaging software (NNT Viewer). The gingival thickness was measured with an endodontic file.

During the procedure, patients were requested to rate their degree of discomfort, postoperative pain, and swelling using a visual analog scale ranging from none (0) to severe (10).

All clinical measurements were recorded at baseline, three months, and six months after the procedure by a single, blinded, qualified investigator. Radiological measurements were performed by a blinded oral and maxillofacial radiologist preoperatively, as well as three months and six months postoperatively. Ridge width was measured at 10.5 mm above the inferior alveolar canal in the coronal plain, and bone density was measured in HU. Scaling and root planning were performed for all patients two weeks before the procedure.

### 2.3. Surgical Procedure

The area was anesthetized with lidocaine 2%, via a nerve block or infiltration. Disinfection with povidone-iodine was done followed by isolation of the area.

#### 2.3.1. Minimalistic Alveolar Ridge Augmentation Technique

A vertical incision of approximately 10 mm was performed mesial to the defect, away from the marginal gingiva and towards the MGJ using Bard-Parker #15 blade (Figure 2B). The vertical incision was placed at right angles to the alveolar bone on the buccal or facial aspect, touching the bone. A subperiosteal pouch was created through the vertical incision across the entire width of the defect with a TKN2 microsurgical stainless steel instrument (Figure 2C).

The blood flowing out of the pouch was collected with a 5 mL syringe in order to manipulate the graft material to prepare the osseous coagulum. The graft material (G Graft^TM^, Surgiwear, Shahjahanpur, India) was then mixed with the blood to prepare an osseous coagulum. The prepared osseous coagulum was loaded into the stainless steel bone graft carrier with an orifice diameter of 7 mm. The material was then gently deposited into the subperiosteal pouch. The size of the pouch orifice was increased in case the graft carrier encountered difficulty entering the poucy. After depositing the graft, gentle digital pressure was applied above the pouch to ensure the adequate fit of the material within the pouch and to avoid the presence of any dead spaces.

After the defect was adequately filled with the graft material (Figure 2F), the wound was closed with simple interrupted silk sutures (Figure 2G).

#### 2.3.2. LLLT Test Sites

At the test sites, LLLT was performed prior to bone graft deposition. A diode laser was employed to deliver LLLT on the exposed bone surface inside the pouch (Figure 2D) prior to grafting. A Fox Arc soft tissue diode laser, with GaAlAs as the active medium and a wavelength of 810 Nm, was used for this purpose. The diameter of the fiber tip used to deliver LLLT was 300 microns. After hemostasis was achieved, the diode laser tip was gently inserted into the pouch. The tip was allowed to touch the bone surface, and irradiation was performed (Figure 2D).

The length of the edentulous ridge was taken into consideration in order to determine the amount of laser radiation to be delivered. The parameters used were 100 Mw, with a maximum energy distribution of 6 J/cm^2^ in continuous wave mode for 60 s per point.

### 2.4. Statistical Analysis

The clinical and radiological measurements were transferred into Microsoft Excel datasheets, and statistical analyses were carried out. All statistical tests were conducted using Statistical Package for Social Sciences (SPSS) version 20.0 (SPSS Inc., Chicago, IL, USA) software. The data’s normality was assessed using the Shapiro–Wilk test. 

If the *p*-value of the Shapiro–Wilk test was >0.05, then the data were considered to be normal. If the value was <0.05, then the data deviated significantly from a normal distribution. Before the beginning of the study, calculations were performed for the power (80%). All quantitative variables are presented as mean and standard deviation. The Shapiro–Wilk test was used to test data normality. Except for the surgical time taken, the *p*-value of the Shapiro–Wilk test was >0.05, indicating that the data were distributed normally. The correlation was evaluated using the paired and unpaired *t*-test, as well as Mann–Whitney U tests. The *p*-value < 0.05 was considered statistically significant.

## 3. Results

Of the 20 patients (*n* = 20) enrolled in the study, all of them underwent the procedure and received follow-up evaluations at one week, three months, and six months (Figure 2H,I) after the procedure, and these assessments revealed no major postoperative complications.

Data was analyzed, and statistical analyses were performed. The majority of the patients treated were males (*n* = 17). The mean age of the subjects was 44 in both study arms; 80% of the sites treated were mandibular posteriors (Table 1).

### 3.1. Clinical Observation

The surgical procedure took less than one hour in both of the groups. At three months follow-up, all of the sites had healed completely, with graft bulk palpable in the treated area. At six months follow-up, all the sites manifested good visible alveolar ridge width and aesthetic harmony. 

A total of 70% of the subjects in the test group and 80% of the subjects in the control group reported no discomfort or swelling during the procedure and postoperative time period. A total of 20% of the cases treated under the test category and 10% in the control group reported mild discomfort. Moderate discomfort was indicated by 10% of the subjects in both groups (Table 2). No patients reported any severe discomfort. One patient presented with exposure of the graft material following removal of the sutures on the fourth day. 

### 3.2. Radiological Findings

Radiographic assessment with CBCT at six months post-surgery showed an increase in MRW for all the cases treated (Figure 3C,D, Table 3). An unpaired *t*-test was performed to compare the difference in gain in the ridge width (GRW) in both groups. The mean GRW was 1.46 ± 0.95 in the test group and 2.14 ± 1.65 in the control group. There was no statistical difference observed in the GRW between the groups. Intra-group comparison of MRW using a paired *t*-test was performed for both groups. The MRW from baseline to six months was 5.05 ± 1.43 to 6.51 ± 1.57 for the test group (*p* = 0.001) and 5.03 ± 0.82 to 7.17 ± 1.98 for the control group (*p* = 0.003), indicating a statistical significance.

The MBD for the test group and control group, at baseline and six months, was 1040.70 ± 490 to 904.30 ± 402.55 and 1138.20 ± 342.50 to 1093.9 ± 380.36, respectively. 

MBD was found to decrease at six months in the grafted sites; however, there was no statistical difference noted between the groups (1040.70 ± 490 reduced to 904.30 ± 402.55, and 1138.20 ± 342.50 reduced to 1093.9 ± 380.36 in the test and control groups at baseline and six months, respectively). CBD was compared for both groups using the Mann–Whitney U test. The mean CBD for the test group was 136.40 and for the control group, it was 44.30, which was not statistically significant.

## 4. Discussion

Most dental surgeries, such as guided bone regeneration and implant surgeries, require extensive soft and hard tissue manipulation, resulting in subsequent loss of the tissue volume. Over the years, surgical methods have been refined in order to take advantage of ultra-conservative approaches which cause the least possible sacrifice of the tissues, contributing to superior healing of the surgical sites.

The rapid ridge resorption which follows tooth extraction is an inevitable sequela, making it more challenging to reclaim the lost alveolar bone. Increased complexity and lack of predictability are the main challenges for all ridge augmentation procedures.

Block bone grafting—onlay grafting, ridge expansion, guided bone regeneration with titanium mesh, and distraction osteogenesis—are some of the invasive procedures presenting dubious outcomes, more often than not. The common complications encountered with such procedures are severe postoperative pain and swelling, membrane exposure, wound dehiscence, loss of graft material, and occurring most frequently, the need for additional membrane to secure the bone graft into position.

Minimally invasive surgical procedures have been revolutionizing surgical periodontology. Its entry into ridge augmentation procedures is relatively recent, with the introduction of laparoscopic procedures [10]. The present study intended to ascertain the efficiency of novel minimalistic alveolar ridge augmentation, in conjunction with low-level laser therapy (MRA+LLLT), in the horizontal reconstruction of the residual alveolar ridge. To optimize the accuracy of the assessment, Cone-beam Computed Tomography (CBCT) was used.

The mean ridge width (MRW) at the baseline in the test group was 5.05 ± 1.43 mm, and for the control group, it was 5.03 ± 0.82 mm. At 6 months, the MRW was 6.51 ± 1.57 mm and 7.17 ± 1.98 mm for the test and control groups, respectively. Thus, the gain in ridge width was 1.40 ± 0.95 mm and 2.14 ± 1.65 mm for the test and control groups, respectively, which showed the effectiveness of the MRA technique for ridge augmentation. The control group did show a marginally higher gain compared to the test group. However, this gain did not reach a statistically significant level. This could be explained by confounding factors, such as the age of the patients and individual healing characteristics.

The horizontal ridge gain achieved in this study was similar to that obtained in a study by Kfir et al. in 2007 (Gain = 1.3–3.9 mm) and Block and Degan in 2004 [5,21]. However, the former study utilized a sequential ballooning technique, with a particulate bone graft and PRF to augment the defect. 

Unlike the results of the current study, a study by Lee et al. [10] showed an average gain of 5.11 mm of ridge width, which could be due to the usage of an inorganic bovine xenograft, in combination with Rh PDGF.

Mean bone density is an important parameter for assessing the quality of regenerated bone. The test site showed an MBD of 1040 ± 490 HU at baseline, which reduced to 904 ± 402.55 HU at 6 months. At the control site, it was 1138.20 ± 342.50 HU, which reduced to 1093.9 ± 380.36 HU from baseline to 6 months. The change in bone density (CBD) was −136.40 HU in the test and −44.30 HU in the control group.

The reduction in bone density (BD) at both sites could be attributed to the ongoing bone remodeling process. There were no comparative studies for the assessment of bone density in minimally invasive surgical procedures; however, in a study by Lorenz et al. [4], bone density was analyzed in a sinus lift procedure. The CBD measured in this study was 500 HU at baseline and 1000 to 1500 HU at 6 months when Nanobone and BioOss, which show similar properties to those of the graft used in our study, were used as graft materials.

The bone substitute employed in the study was G Graft (combination of HA and collagen). G graft showed better results when used for socket preservation in molar extraction sockets compared with G Bone (a combination of BTCP and HA) in a study by Panday V et al. in 2015 [22].

HA has been used as a graft substitute in regenerative periodontics for decades. A study by Lindgren et al. in 2010 [23] showed improved results for inorganic bovine HA when used for alveolar bone defect regeneration, which is in concordance with the results of this study, demonstrating the role of HA as a scaffold material for alveolar ridge augmentation. 

Comparison between the test and the control groups concerning the gain in ridge width, as well as bone density, did not reveal any statistically significant difference (*p* = 0.27 and 0.29). However, the intra-group comparison of the gain in ridge width (GRW) showed a statistically significant difference from baseline to six months. 

The intragroup comparison of bone density (BD) did not show a significant change in the test or control groups. The control group’s mean bone density (MBD) decreased from 1138.20 342.50 HU to 1093.90 380.36 HU, while the test group’s MBD decreased from 1040.70 490.74 HU to 904.30 402.35 HU at baseline and six months. There seems to be no peripheral effect of biostimulation on bone healing in terms of regeneration. 

There have been a handful of studies regarding the impact of laser biostimulation on bone regeneration in human subjects. A study by Garcia et al. in 2015 [24] used a diode laser with similar parameters to improve osseointegration around implants, but this study did not show any advantages similar to those obtained in our study.

On the other hand, a study by Mandic in 2015 [25] evaluated bone healing after LLLT in immediate implants and found that LLLT promotes bone healing around implants, in contrast to the results of our study. However, he performed laser irradiation for 7 consecutive days, whereas in the present study, LLLT was only performed for a single day. 

The variations in the results observed for LLLT could be explained by the role of individual laser parameters, namely the active medium, wavelength, energy, exposure time, and frequency of exposure.

The mean surgical time required for the test group was 0.45 ± 0.35 h, and for the control group, it was 0.61 ± 0.37 h. (*p* = 0.36). The difference between the groups was not statistically significant; however, it can be explained by the fact that the control group exhibited a relatively long span of edentulous sites.

We found the present technique to be simple, with reduced patient morbidity and a relatively reduced risk of loss of augmentation volume. Patient-reported outcomes are an integral part of the quality of treatment. In both the test and control groups, specific outcomes regarding any discomfort or swelling intraoperatively and postoperatively were recorded. A total of 70% of the subjects in the test group and 80% of the subjects in the control group reported no discomfort or swelling during the procedure and postoperative time period, while 10% of the subjects reported moderate discomfort, and the remaining fraction of the patients reported only mild discomfort. The difference in patient outcomes between the two groups was not statistically significant (*p*-value = 0.45). 

Early in the1980s, Kent et al. proposed minimally invasive horizontal ridge augmentation (MIHRA) utilizing a subperiosteal tunneling technique in which a small vertical incision was made in the alveolar ridge, and hydroxyapatite particles were administered underneath the periosteum. Initially, the graft was successful, but studies conducted in the late 1980s revealed that injected hydroxyapatite particles were unstable, and fibrous encapsulation was observed, resulting in partial bone formation [12]. Since then, MIHRA has been has been scarcely studied. In 2017, Lee et al. successfully employed the subperiosteal minimally invasive aesthetic ridge augmentation technique (SMART) for the reconstruction of the alveolar ridge [10].

Our study effectively employed a similar technique for maxilla and mandible ridge reconstruction. Based on the literature, this is the first study to investigate the bone quality (in Hounsfield units) regenerated by CBCT for minimally invasive ridge augmentation. The technique not only yielded significant ridge gain, but also showed better patient acceptance and simplicity of execution. This clinical study is unique in that it also explores the adjunctive utilization of LLLT for bone regeneration in human subjects.

This study has several limitations. First, the sample size was small; thus, a larger sample size and a longer follow-up duration after implant placement could shed more light on the overall outcome of the study. Secondly, laser irradiation could have been performed for more than one day, which may have improved the outcomes in the test group. Additionally, other growth-promoting materials, in combination with bone grafts, could have been employed that may have altered the outcomes.

## 5. Conclusions

We conclude that the MRA technique using G graft may be a simple and minimally invasive alternative to other ridge augmentation procedures. In this study, we observed no significant impact of LLLT in bone regeneration.

## Figures and Tables

**Figure 1 medicina-59-01178-f001:**
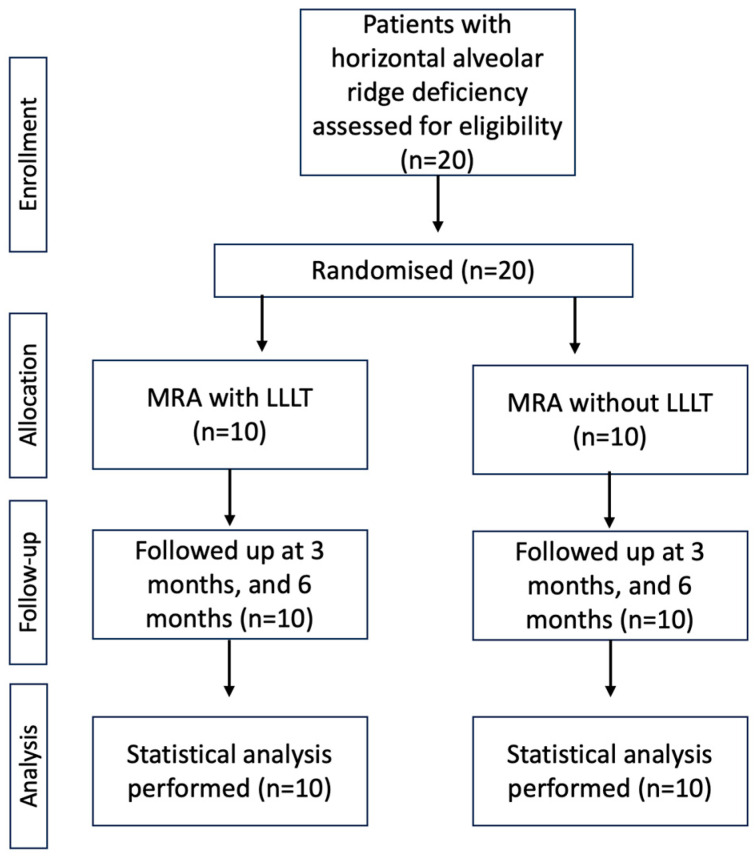
Consort flow diagram showing the randomization, follow-up, and analysis involved in the study.

**Figure 2 medicina-59-01178-f002:**
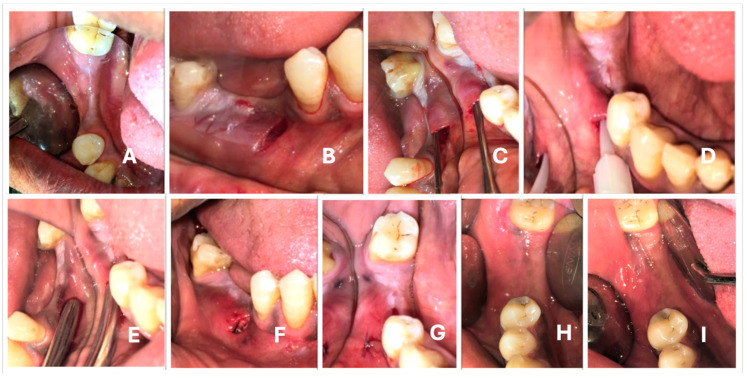
Surgical technique: (**A**) preoperative ridge width; (**B**) vertical incision on the attached gingiva mesial to the defect; (**C**) flap elevation with periosteal elevator; (**D**) diode laser tip inserted into the pouch for LLLT; (**E**) graft carrier inserted for bone graft deposition; (**F**) graft filled; (**G**) Vicryl sutures placed; (**H**) three months follow-up; (**I**) six months follow-up.

**Figure 3 medicina-59-01178-f003:**
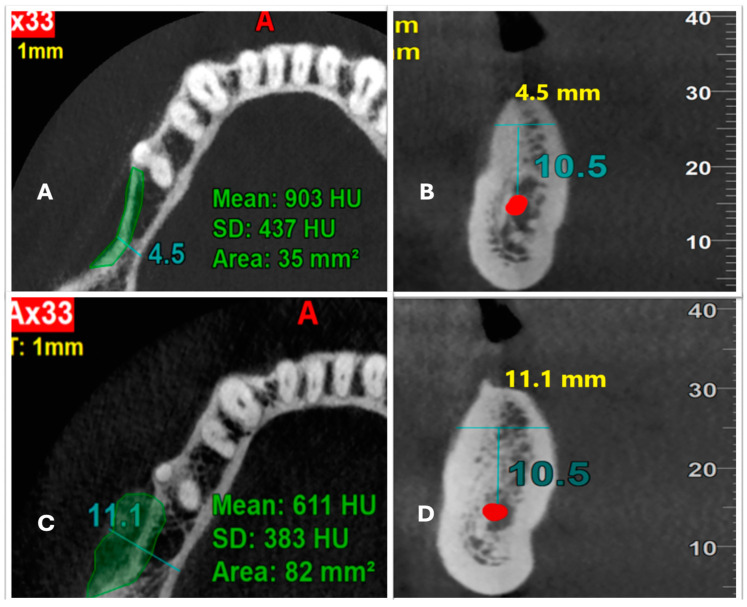
CBCT analysis of the ridge width and bone density. (**A**) Preoperative ridge width and bone density in the axial plane; (**B**) preoperative ridge width shown in the coronal plane 10.5 mm above the inferior alveolar nerve; (**C**) postoperative ridge width and bone density in the axial plane; (**D**) postoperative ridge width shown in the coronal plane 10.5 mm above the inferior alveolar nerve.

**Table 1 medicina-59-01178-t001:** Description of patients and graft sites.

Description of Patients and Graft Sites		
Patients	Number	Percent
Total	20	100
Sex (Male	17	85
Sex (Female)	3	15
**Graft sites**		
Maxillary anterior	0	0
Maxillary posterior	4	20
Mandibular anterior	0	0
Mandibular posterior	16	80

**Table 2 medicina-59-01178-t002:** Patient reported outcomes for both the control and test groups.

Groups	Patient Reported Outcomes			*p* Value
	None	Mild Pain	Moderate Pain	
**Test**	7	2	1	0.81
	70.0%	20.0%	10.0%	
**Control**	8	1	1	
	80.0%	10.0%	10.0%	

**Table 3 medicina-59-01178-t003:** Radiographic measurements.

	Test Group			Control Group		
	Baseline	6 Months	*p* Value	Baseline	6 Months	*p* Value
**Mean Ridge Width (MRW) in mm**	5.05 ± 1.43	6.51 ± 1.57	0.001 *	5.03 ± 0.82	7.17 ± 1.98	0.003 *
**Mean Bone Density (MBD) in HU**	1040.70 ± 490	904.30 ± 402.55	0.101	1138.20 ± 342.50	1093.9 ± 380.36	0.45
**Mean Gain in Ridge Width (GRW) in mm**	1.46 ± 0.95			2.14 ± 1.65		0.27
**Change in Bone Density (CBD) in HU**	−136.40 ± 236.08			−44.30 ± 180.89		0.29

* Significant according to the unpaired *t* test.

## Data Availability

The dataset used in the current study is available upon reasonable request.

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
