# Peer review of "Alveolar Ridge Augmentation Assessment Using a Minimalistic Approach, with and without Low-Level Laser Therapy (LLLT)—A Comparative Clinical Trial"

_medicina, 2023, doi:10.3390/medicina59061178_

Round 1
Reviewer 1 Report
The study about SMART application with and without Low-Level Laser Therapy is good since no studies have been done or reported. However, some points are lacking in describing the finding/result, discussion, and conclusion. Many data are presented as tables but with a very lack of description. In addition, there is a repetition of the data description for the discussion part, which would be much more appropriate to be highlighted with essential points to be discussed. For the conclusion part, their sentence is not conclusive enough to the finding. Other than that, several grammatical errors are noticed, especially in the methodology part. Please refer to the highlighted words/sentences for more details. Overall, English needs moderate editing to be more accessible, read, and understood.

The study about SMART application with and without Low-Level Laser Therapy is good since no studies have been done or reported. However, some points are lacking in describing the finding/result, discussion, and conclusion. Many data are presented as tables but with a very lack of description. In addition, there is a repetition of the data description for the discussion part, which would be much more appropriate to be highlighted with essential points to be discussed. For the conclusion part, their sentence is not conclusive enough to the finding. Other than that, several grammatical errors are noticed, especially in the methodology part. Please refer to the highlighted words/sentences for more details. Overall, English needs moderate editing to be more accessible, read, and understood.
Author Response
We thank you for the time taken to review our manuscript. Please find the reply to the comments attached.

Reviewer 2 Report
This is an interesting study, it is well described and very usful for clinicians.
I only have one comment to the authors:
Fig1 Caption - line186: it is mentioned that: “(B) Vertical incision on the attached gingiva distal to the defect” but in surgical procedure in material and methods line 155, it is mentioned: A vertical incision of approximately 10 mm is placed mesial to the defect. Please make the necessary correction.
Author Response
We are grateful for your review of our manuscript. Please find the response to the comments attached.

Reviewer 3 Report
A power study to determine level of significance and presentation of a statistical analysis to determine the minimal sample size also needs to be done. I do not think that 10 subjects in the experimental and control groups is sufficient to provide any substantive data to base conclusions on.
Furthermore, additional methods for determining amount of bone augmentation should be included - perhaps direct observation?
In addition, there are English language grammatical errors that need correction.
There exist multiple grammatical English language errors that require addressing.
Author Response
Please find the response to the comments attached and the changes are made accordingly in the manuscript.
Regards

Reviewer 4 Report
This manuscript presents the results of a RCT conducted to investigate a minimal invasive approach on horizontal ridge augmentation.
The topic is of relevance in the field of oral surgery and clinical data is always worth a publication. Nevertheless, the presentation of this investigation is not suitable for a publication in the current state.
Major concers:
1. All clinical studies should follow the CONSORT statement for reporting clinical trails. Please stick close to this recommendations.
2. The inclusion criteria are not described clearly. Please provide more detailed information on the size of bone defects included in this study.
3. Statistical analysis: should be part of Materials and Methods. Please revise the whole paragraph. If this manuscript would have undergone a thourough revision by the authours, the repetetive information in this section would have been noticed.
At this point, I stop the detailed review of the manuscript. I encourage you to revise all parts of the work, according to the guidelines for such papers. I recommend to look for a recently published clinical study in this area of interest and stick close to the form of the presentation (tables, figures, statistics). Please also provide a ciritical discussion of your methods and results.
The manuscript needs s thourough revision of the language (by a native speaker). There are many phrases which are confusing and very hard to understand.
Author Response
Kindly find attached the response to the comments and the changes are done accordingly in the manuscript.
Regards.

Round 2
Reviewer 4 Report
Thank you for the revision of the manuscript according to the referees suggestions. One minor suggestion: Why not add a flowchart of the study design (showing both arms of the study), to provide an easier access for the reader?
Author Response
We thank you for the review. As suggested, a consort flow chart is added in the methodology section.
Best regards,
Dr Shaeesta